# Magnetic Compression Anastomosis–Past Experience and Current Proposals for Further Development in Pediatric Minimally Invasive Surgery

**DOI:** 10.3390/children10081328

**Published:** 2023-08-01

**Authors:** Anatole M. Kotlovsky, Oliver J. Muensterer, Vasily V. Nikolaev, Alexander Y. Razumovskiy

**Affiliations:** 1Department of Pediatric Surgery, Dr. von Hauner Children’s Hospital, Ludwig-Maximilians-University Medical Center, Lindwurmstrasse, 480337 Munich, Germany; ank424@gmail.com; 2Department of Pediatric Surgery, N.I. Pirogov Russian National Research Medical University, Russian Children’s Hospital, Leninsky Prosp 117, 119571 Moscow, Russia; vasnik@yandex.ru (V.V.N.); 1595105@mail.ru (A.Y.R.); 3Department of Pediatric Surgery, Pirogov Russian National Research Medical University, Filatov Children’s Clinical Hospital, Ulitsa Sadovo-Kudrinskay 13, 123001 Moscow, Russia

**Keywords:** magnetic compression anastomosis, magnamosis, minimally invasive surgery, pediatric surgery, children

## Abstract

Originating in the 1970s, magnetic compression anastomosis (MCA) has lately been revisited with a focus on minimal invasive surgery (MIS). The aim of this report is to reappraise our earlier experience with MCA with the intention of facilitating future MCA advancement. A retrospective review was conducted regarding preclinical experiments and clinical trials at a single institution from 1980 to 1995. The reviewed information was compiled and appraised to generate proposals for future MCA use. The experimental studies, including 250 MCA cases in gastrointestinal and urinary tract animal models, demonstrated the technical versatility of MCA as well as the superior biomechanical characteristics in comparison to hand-sewn anastomoses. Clinical trials encompassed 87 MCA procedures in 86 children, 2 to 10 years of age, involving the following techniques: non-operative esophageal recanalization (*n* = 15), non-operative ileostomy undiversion (*n* = 46), Swenson pull-through (*n* = 10), non-operative urethral recanalization (*n* = 5), and extravesical ureterocystoneostomy (*n* = 11). Clinical MCA was found to be successful in over 87% of cases. MCA limitations concerning anastomotic failure and scarring were thought to be mostly due to inadequate magnetic compression. Based on our historic experience, we propose further research on the technical aspects of MCA, along with the biological aspects of anastomotic tissue remodeling. Magnets should be designed and manufactured for a wide spectrum of pediatric surgical indications, particularly in combination with novel MIS techniques.

## 1. Introduction

First introduced in adult colorectal surgery in the Netherlands some 40 years ago [1], magnetic compression anastomosis (MCA) has lately been revisited under the term “magnamosis”, particularly in the context of pediatric minimally invasive surgery (MIS) [2,3,4,5,6].

Remarkably, the number of publications and presentations on the topic has been increasing over the past decade, especially in pediatric surgery. While this is indicative of a greater interest in the potential use of MCA, the more recent clinical applications are still limited to case presentations and small patient series [4,5,6,7,8,9,10,11,12,13].

Following the historical Dutch report [1], a pilot project on MCA use in pediatric surgery, including pre-clinical animal experiments and clinical trials, was implemented at the Department of Pediatric Surgery, N.I. Pirogov Russian National Research Medical University the Moscow, N.F. Filatov Children’s Hospital, Moscow, starting from the 1980s and continuing well into the early 1990s. Most of these data, however, have not been published in the English literature.

After the described experiments, magnetic compression anastomosis was not pursued for various reasons, but it experienced a renaissance recently. The goal of this article is to bridge the knowledge gap between the historic experience and current studies on magnetic compression anastomosis. Therefore, the aim of this report is to reappraise our earlier and relatively extensive experience with MCA, and to evaluate the results that may be pertinent to contemporary proposals for MCA advancement.

## 2. Materials and Methods

A retrospective review of the above-mentioned single-institution materials, dated from 1980 to 1995, was conducted on MCA in preclinical experiments on animal models and subsequent clinical trials. The originally designed MCA techniques for specific pediatric surgical and urological applications were assessed from historic records. The sources of the reviewed information included literature publications, summaries of PhD theses, certified patents documentation (all in Russian language), and some personal records of the authors.

Specifically, the analysis of the information focused on the aspects of the safety and technical feasibility of MCA, considering the biological formation and biomechanical characteristics of the newly formed anastomoses with a focus on potential prospective and innovative clinical applications.

The extrapolated information was used to propose ideas for MCA advancement in pediatric surgery in accord with recent reports on contemporary applications of MCA, as identified in a literature search.

## 3. Results

### 3.1. General Aspects and Preclinical Experiments

There was a total of 250 preclinical, experimental MCA cases of different types in various locations in the digestive and urinary tracts, primarily performed in canine and rabbit models (Table 1). All experiments were performed with the use of Samarium Cobalt (SaCo) magnets with a variable geometrical shape, size, and coercivity force.

Technical and biomechanical characteristics were evaluated in comparison with corresponding hand-sewn anastomoses as controls in animal models [14,15,16,17,18,19,20,21,22,23].

Performing MCA was simple, effectively facilitating the formation of sutureless anastomoses via an endoluminal approach [14,15,16,17,18,19,20,21].

The results of biomechanical and histomorphological studies on all animal models showed that, at various postoperative intervals from 1 day to 1 month, the MCA in general had characteristically superior anastomotic properties compared to the hand-sewn controls (Figure 1).

In particular, MCA demonstrated high tensile strength, most likely due to the healing by primary intention with no resultant leakage or excessive scarring or strictures [16,17,18,19,20,21].

Tensile strength tests on the intestinal rabbit models at 1, 5, 7–10, 14, and 21 postoperative days demonstrated bursting pressures that were 1.5 to 2-fold greater than after hand-sewn anastomoses in a control series (*n* = 35) [15,16].

Histological examinations of the MCA at 1, 5, 7–10, 14, 21, and 30 days postoperatively confirmed seamless healing in both the gastrointestinal (*n* = 96) and urinary (*n* = 84) tract models. We also found primary epithelization within 7–10 days (Figure 1B). In contrast, healing of hand-sewn analogues in all control groups was associated with ongoing suppurative inflammation with intramural micro-abscesses around the absorbable suture material (polyglactin or catgut), which were identifiable up to 1 month postoperatively, and resulted in fibrous tissue proliferation (Figure 1C) [15,17,18,20].

In seven cases of experimental colorectal MCA, anastomotic perforation and dehiscence was noted when using magnets with high coercivity force and small compression surfaces. A combination of these mechanical factors led to focal tissue overload with subsequent tissue tearing and perforation [15]. Therefore, a limitation for MCA was postulated as using strong magnets with minimally rounded edges and small compression surface.

In summary, the following factors were identified for successful MCA formation.

At the anastomotic line, epithelization and connective tissue matrix proliferation dynamically develop at the so called “zone of moderate compression”, where the compression forces decrease gradually towards the periphery. This is effectively modulated by the rounded edges of magnetic compression surfaces. Flat magnets should not be used for MCA because they are associated with a higher degree of scarring and, thus, stricture formation.In the demarcated central zone, maximal compression is required to ensure reliable anastomotic coaptation, and to exponentially compress the subjected tissue to the point of desiccation and necrosis, preventing suppurative necrosis with inflammation within the anastomosis itself, which may lead to inadequate anastomosis formation and increased proliferation of granulation tissue.At the demarcation zone that constitutes the inner anastomotic line, epithelization steadily progresses and bridges the rim of the thinned desiccated central tissue. The magnet thereby gradually becomes detached and passes distally within 7 to 10 days; thus, the anastomotic healing by primary intention is completed (Figure 2) [15,17,18,20].

### 3.2. Review of Clinical Trials

A review of the records revealed a total of 87 MCA procedures performed in 86 selected patients, 2 to 10 years of age. The MCA clinical applications involved using SaCo magnets of specific geometric configuration, size, and compressing force, based on the preclinical experimental evaluations detailed above. Indications comprised various pathological conditions of the gastrointestinal and urinary tracts in pediatric surgical practice [15,16,17,18,19,22,23,24], detailed as follows.

#### 3.2.1. Non-Operative MCA Esophageal Recanalization

The original technique was developed to treat short-length esophageal strictures using magnetic cylinders by means of compressing and ablating the underlying scar tissue (Figure 3). The technique was applied in fifteen patients (*n* = 15) with esophageal strictures 3 to 15 mm in length [14,16,22,23].

Under rigid esophagoscopy, specially designed magnetic cylinders 10, 12, or 14 mm in diameter were deployed to the stricture site via the gastrostomy and perorally over a guide string, which was passed through the stricture narrowing after previous bougienage. Under fluoroscopy, the magnets were then brought together into coaptation.

The magnets were followed by daily repeat X-rays for the next 3 to 5 days. The resultant MCA formation usually occurred within 7 days, when the magnets detached from the anastomosis. They were subsequently removed under fluoroscopic control and endoscopic assistance by retracting them by the attached strings [22,23].

Complete esophageal recanalization was achieved in nine cases, five of which were congeital membranous stenoses and four of which were postoperative anastomotic strictures. Adverse outcomes included esophageal perforation in one case and esophageal re-stenosis in five [22,23].

#### 3.2.2. Non-Operative MCA Ileostomy Undiversion

A technique for enterostomy undiversion was developed using flexible silicon-coated segmented block-shaped magnets in a side-to-side configuration (Figure 4). The technique was used in forty-six patients with double barrel ileostomy when undiversion was clinically indicated [24].

Technically, the magnets were fashioned with attached strings for easy retrieval after successful MCA formation. They were manually placed into each stomal limb and brought to proper coaptation under fluoroscopic guidance. Subsequent MCA formation usually occurred between 7 and 10 days [24].

Successful MCA enterostomy undiversion was achieved in forty-four patients. There were no cases of leakage, and the intestinal passage was restored in all patients. In two cases, magnet placement was aborted intraoperatively due to gross tissue interposition between the stomal limbs because it was deemed unsafe to bring magnets into the coaptation [24].

#### 3.2.3. Swenson Type MCA-Based Pull-through for Hirschsprung Disease

A modified MCA-based Swenson pull-through technique using flexible silicon-coated segmented magnetic rings of 20 to 40 mm diameter was designed to treat rectosigmoid Hirschsprung disease in 10 patients (Figure 5) [14,15,16].

Technically, the procedure entailed an open approach via conventional laparotomy. The affected bowel segment was conventionally mobilized with deep pelvic dissection to the lower rectal level. The proximal magnetic ring was delivered intra-luminally through the anus to the upper limit of the intended resection, which was followed by eversion of the mobilized bowel portion trans-anally. The distal magnetic ring was then placed over the everted recto-sigmoidal cylinder. The magnetic rings were brought into coaptation at the level above the anal verge, creating an end-to-end colorectal MCA. A rectal tube was inserted intra-luminally through the magnetic rings proximally to the anastomosis and the everted bowel cylinder was then excised.

Magnet coaptation was verified intraoperatively using fluoroscopy and monitored postoperatively by serial radiographs. The MCA usually formed within 8 to 12 days. Once the anastomosis had formed, the magnets detached and were manually removed through the anus [14,15,16].

An intact colorectal MCA with no leakage or subsequent stenosis was achieved in six patients. Adverse outcomes included post-anastomotic stenosis in two patients (*n* = 2) with bulky-thickness bowel wall interposed between the magnetic rings [16].

#### 3.2.4. Non-Operative Urethral Recanalization

Small diameter magnetic cylinders were used to treat short-length post-traumatic urethral strictures (Figure 6). The technique was applied in five patients with short strictures that were less than 5 mm in length [17,19,25].

The upper magnetic cylinder of 4 to 6 mm diameter was delivered via rigid cystoscopy through a temporary cystostomy. The lower magnetic cylinder was delivered via cystoscopy through the meatus. Both magnets were introduced over a guide string that had been previously placed across the stricture. Fluoroscopy was used to verify good coaptation of the magnets.

Urethral MCA formation usually occurred within 7 to 10 days. After anastomosis formation, the magnets were removed by retracting them by the attached string using cystoscopy and fluoroscopic control [17,19,25].

Complete restoration of urethral patency and normal voiding patterns were achieved in four patients with membranous type strictures (*n* = 4). Post-anastomotic restenosis was noted in one patient with a more extensive stricture of 5 mm in length [17,19].

#### 3.2.5. Extravesical Ureterocystoneostomy

A technique of ureteric reimplantation using small magnetic rings of 4 to 5 mm in diameter was designed to create an extravesical MCA between the ureter and urinary bladder (Figure 7). It was employed on a total of 11 ureters in 10 patients with grade 3 to 4 vesicoureteral reflux (VUR) [18,26].

The procedure involved an open preperitoneal approach via a Pfannenstiel-type incision. The uretero-vesical segment was mobilized, the distal ureter was resected, and a Lich-Gregoir detrusorotomy was performed in a conventional fashion. The ureterocystoneostomy was accomplished by placing the proximal magnetic ring over a stent into the lumen of the ureter. At the point of intended anastomosis, the distal tip of the stent (with the curl cut off) was brought into the bladder by using an attached stylet piercing through both the ureteric wall and the distal corner of the denuded submucosal surface of the bladder wall. The ureteric lumen was then closed with continuous absorbable sutures. The distal magnetic ring was introduced into the bladder over the distal tip of the stent brought out via a mini-cystotomy, followed by a cystostomy tube placed over the stent. The magnets were then approximated, creating a side-to-side MCA between the full-thickness ureteric wall and the submucosal layer of the bladder wall. Ureterocystoneostomy was completed by closing the tunnel in a Lich-Gregoir fashion using interrupted absorbable sutures.

MCA formation occurred within 7 to 10 days. The magnets detached from the anastomosis once it was formed and were removed via the cystostomy by retracting the stent and cystostomy tube [18,26].

The procedure was successfully completed in all 10 patients. No recurrent VUR or anastomotic strictures were noted in the postoperative follow-up [18].

### 3.3. Clinical Success for MCA

A summary of clinical outcomes is presented in Table 2, demonstrating that across all indications, MCA was successful in over 87% of cases [17,18,19,22,23,24].

### 3.4. Recognized Adverse Effects Associated with MCA

Adverse outcomes encountered in our experience are listed in Table 2. In detail, they comprised:Inability to safely perform ileostomy undiversion in two patients due to interposing of excessively thick tissue, so that the procedure was aborted due to safety concerns [24];Recurrent strictures after esophageal recanalization in five patients with corrosive strictures of irregular shape and more than 10 mm in length [22,23];Recurrent strictures after urethral recanalization in a patient with a stricture approximately 5.0 mm in length (*n* = 1) [17,19];Restenosis after Swenson colorectal MCA in two patients with a bulky bowel wall [15,16] that impeded proper mating of the magnets.

The pathomechanism of postoperative restenosis or stenosis (esophageal and urethral, as well as colorectal) was thought to be related to a relatively insufficient magnetic compression force, which appeared to occur in circumstances of excessive distance between magnets due to disproportionately bulky tissue interposition. Conversely, esophageal perforation was possibly due to forceful compression applied to extensive stricture scar tissue and its inability to exhibit normal healthy tissue healing.

The above scenarios should be considered limitations for clinical MCA applications [15,16,18,20,21,22].

## 4. Discussion

Contrary to what many believe is a recent, novel development, MCA has previously been explored in both animal experiments and clinical trials in our institution in Moscow several decades ago. Our historic experience with MCA, reported for the first time in the English literature, dates as far back as the 1980s and early 1990s. Nonetheless, many of the issues regarding MCA and its challenges resonate with contemporary reports. Our experience spans a large spectrum of indications, ranging from pediatric upper gastrointestinal, colorectal, and urological applications.

Recently, other groups have been reviving MCA for use in various pediatric surgical indications [13], including esophageal atresia repair [11,12,27], rectal atresia [9], duodenal stenosis [10], vascular anastomosis [28], and for re-cannulation of the hypopharynx [29]. Several experimental studies have been conducted in swine [30,31], rabbits, and dogs [32]. The studies confirm the above findings from decades ago and also seem to suggest that magnets can be used for creating the anastomosis, but that simultaneous approximation of pouches or structures under tension will lead to increased rates of anastomotic stricture [12].

In the field of adult surgery, magnets have also been used for a wide spectrum of applications, including duodeno–ileal anastomosis [33,34], ureterostenosis after kidney transplantation [35], hepaticojejunostomy [36,37], biliary strictures [38,39,40], as well as for foregut issues, such as strictures after esophageal cancer surgery [41] and sleeve gastrectomy [42].

Given the retrospective review nature of our historic experience with MCA, we acknowledge that this report lacks a certain grade of evidenced-based robustness. The technical aspects of each individual study are described according to our laboratory notes, scientific reports in Russian language, and doctoral theses that resulted from the experiments and clinical studies. Nonetheless, for those interested, further details and copies of the original materials are available from the corresponding authors upon specific request.

The first-hand information presented here accurately conveys our initial experience on the biological basis and applicability of MCA. To our knowledge, these are some of the first clinical applications reported, and constitute a pioneering effort to employ MCA in pediatric surgery. Particularly, the MCA experience described here were carried out in relatively large numbers, whereas many recent articles report much smaller case numbers of similar techniques.

Considering our earlier experience, we outline the rationale for the potential MCA application in gastro-intestinal and urinary tract surgery on the grounds of its superior biomechanical properties versus hand-sewn anastomosis and technical versatility allowing a greater scope for creating anastomoses, especially, in settings of the non-operative endoluminal approach and challenging surgical access in difficult anatomical areas.

The reasons for using MCA rather than a handsewn anastomosis include purely endoscopic applicability and technical simplicity, which may decrease operative times. These aspects are particularly important in multimorbid infants, which do not tolerate long operative times. In addition, MCA produces an even circular anastomosis, with less variability compared to the variability of individually placed sutures. The drawback of MCA is the timeframe in which the anastomosis forms. Compared to a hand-sewn anastomosis, the MCA is not immediately patent after placing the magnets but needs 7 to 10 days to mature. This delayed patency may postpone enteral feedings for several days, compared to a hand-sewn anastomosis. Nevertheless, we believe that MCA should be considered for minimally invasive applications in small infants that do not tolerate long operative times.

We hope that this information will provide a stimulus for the current working groups on MCA to incorporate our findings and ideas for relevant future innovative development in pediatric surgery, in particular, in conjunction with a MIS approach. Combining MCA and MIS techniques create synergy and seem like a worthwhile future direction.

At this point, we propose that further research and development of MCA is undertaken to rationalize its formation, confirm efficacy for the described indications, and explore new fields of application, particularly in pediatric surgery. Optimally, these endeavors should be conducted with interdisciplinary medical, bioscientific, and engineering input. Some of the work packages we propose are:Designing and industrially manufacturing biocompatible and safe rare earth magnetic devices for specific anastomotic indications with optimal parameters of compression, size, and specific geometrical shape for creating MCA in pediatric gastrointestinal and urinary structures.Designing and industrially manufacturing an auxiliary magnetic driving device with computerized technology that would facilitate intraluminal magnet positioning and coaptation, ensuring efficient MCA creation while electronically monitoring and securing optimal tissue compression. This should be facilitated by measuring the distance and the resulting effective magnetic coercivity force, thereby detecting possible undue anastomotic tissue tension and/or bulky tissue interposition between magnets in real-time.Push for further research into the development and standardization of MCA in conjuncture with pediatric MIS with a focus on the following procedures:▪Endoluminal recanalization in short-length obstructive lesions with a various pathogenesis and localization;▪Non-operative undiversion of intestinal stomas;▪Laparoscopic pull-through;▪Laparoscopic extravesical ureteric reimplantation;▪Laparoscopic biliary-digestive reconstruction;▪Laparoscopic duodenal atresia repair;▪Laparoscopic-assisted repair of certain anorectal malformations;▪Thoracoscopic esophageal atresia repair with and without a fistula.

Our proposals for designing and industrially manufacturing magnets as a specific anastomosing device along with an auxiliary computerized magnetic driving device are intended to standardize MCA technical performance and to ensure its optimal biological formation while avoiding risks of the MCA limitations, as identified in our past experience. We feel that this is paramount to ensure safe and effective MCA for neonatal and older pediatric patients. In summary, following our past experience with MCA, we believe that future research on MCA should be combined with pediatric MIS techniques that were not available when we performed our experiments and clinical studies.

## 5. Conclusions

This is the first comprehensive report of our historic MCA experience in the English language. The experience described in this report can be considered a true pioneering effort, which unfortunately has not been as visible as it merits so far. Our studies show that MCA should be considered as a valid alternative to hand-sewn anastomoses in a wide spectrum of pediatric surgical problems, particularly for future MIS applications. Our report underlines the surgical versatility of MCA, particularly for endoluminal applications using endoscopic delivery techniques. It also shows the importance of finding the optimal compression pressure and other biophysical properties, with the goal of producing high-quality anastomoses. Based on our findings, novel devices should be designed for specific indications, taking into consideration biological healing of the particular tissues and the required magnetic force to form the anastomosis.

## Figures and Tables

**Figure 1 children-10-01328-f001:**
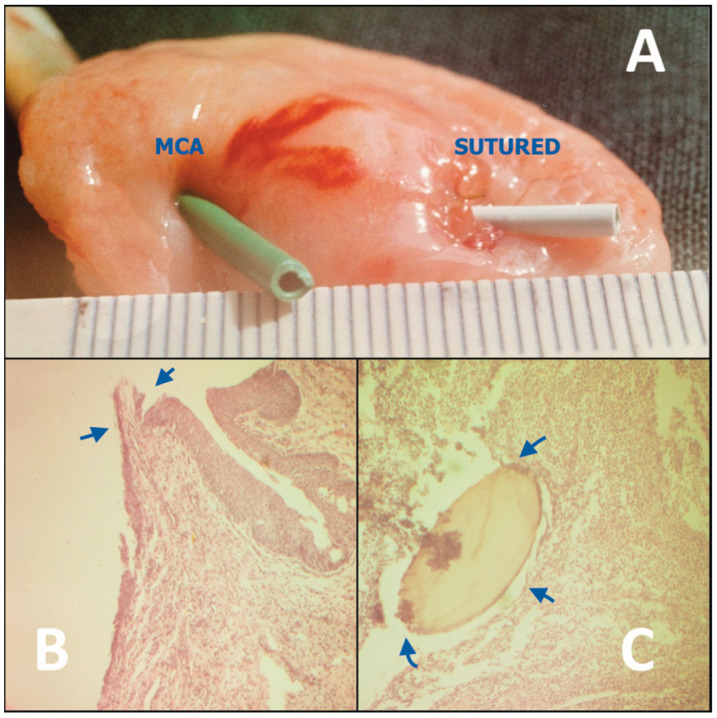
Canine model of an MCA constructed between the ureter and urinary bladder (**left**) in comparison with a hand-sewn control (**right**), shown on postoperative day 14. (**A**) Compared to the hand-sewn anastomosis, which shows swelling and granulation tissue, the MCA more closely resembles the native ureteric orifice. (**B**) Histological image of the MCA, showing complete epithelization at the compression/anastomotic line (arrows). (**C**) Histological image of the sutured anastomosis with intramural suppurative changes around the suture material (arrows).

**Figure 2 children-10-01328-f002:**
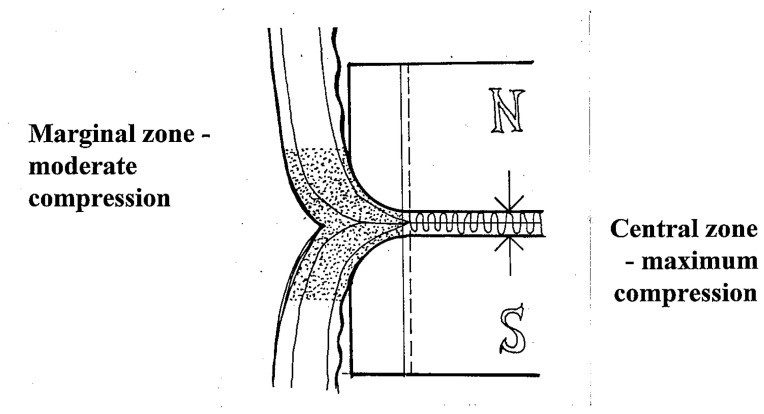
Diagram of optimal MCA formation. The central tissues are highly compressed to produce desiccation and necrosis, while gradient compression in the periphery allows healing and promotes trans-anastomotic mucosal bridging (N: magnetic north, S: magnetic south).

**Figure 3 children-10-01328-f003:**
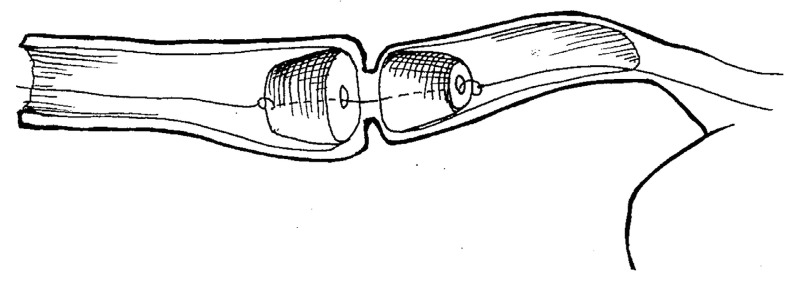
Schematic principle of non-operative endoluminal esophageal recanalization of strictures using magnetic cylinders.

**Figure 4 children-10-01328-f004:**
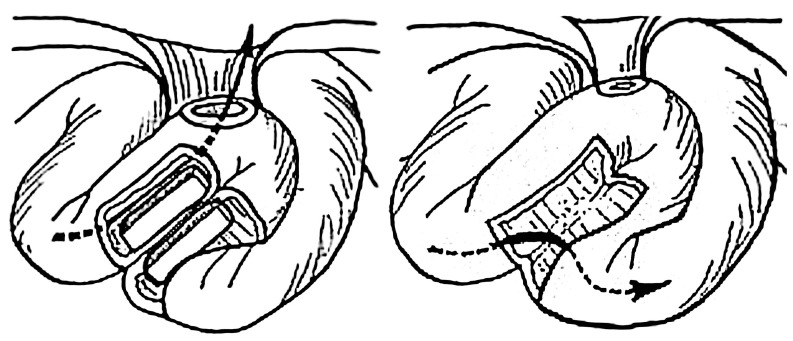
Schematic drawing of endoluminal loop ileostomy undiversion using magnetic blocks. The arrow marks the flow of stool.

**Figure 5 children-10-01328-f005:**
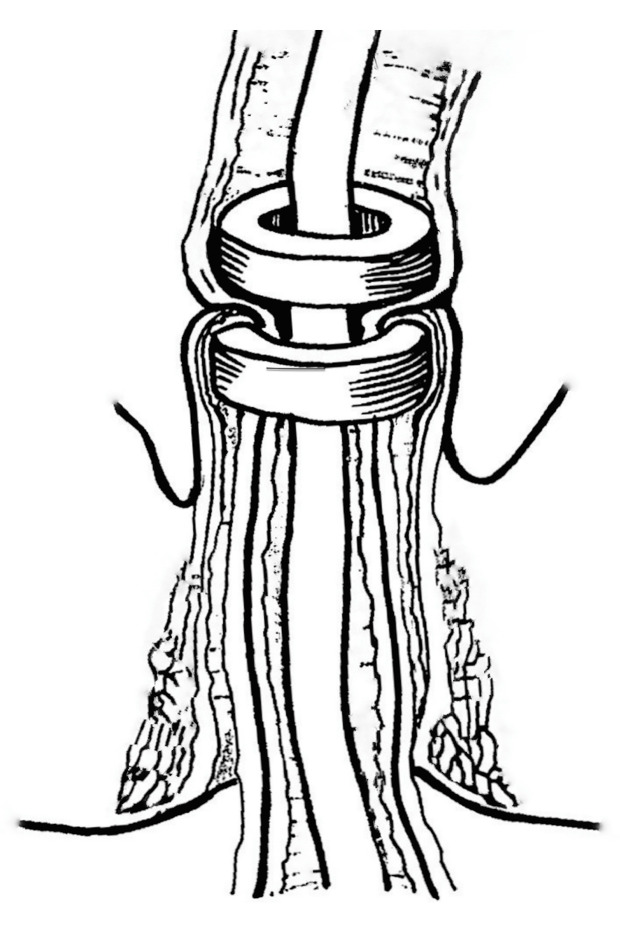
Principle of MCA-based Swenson pull-through for Hirschsprung disease.

**Figure 6 children-10-01328-f006:**
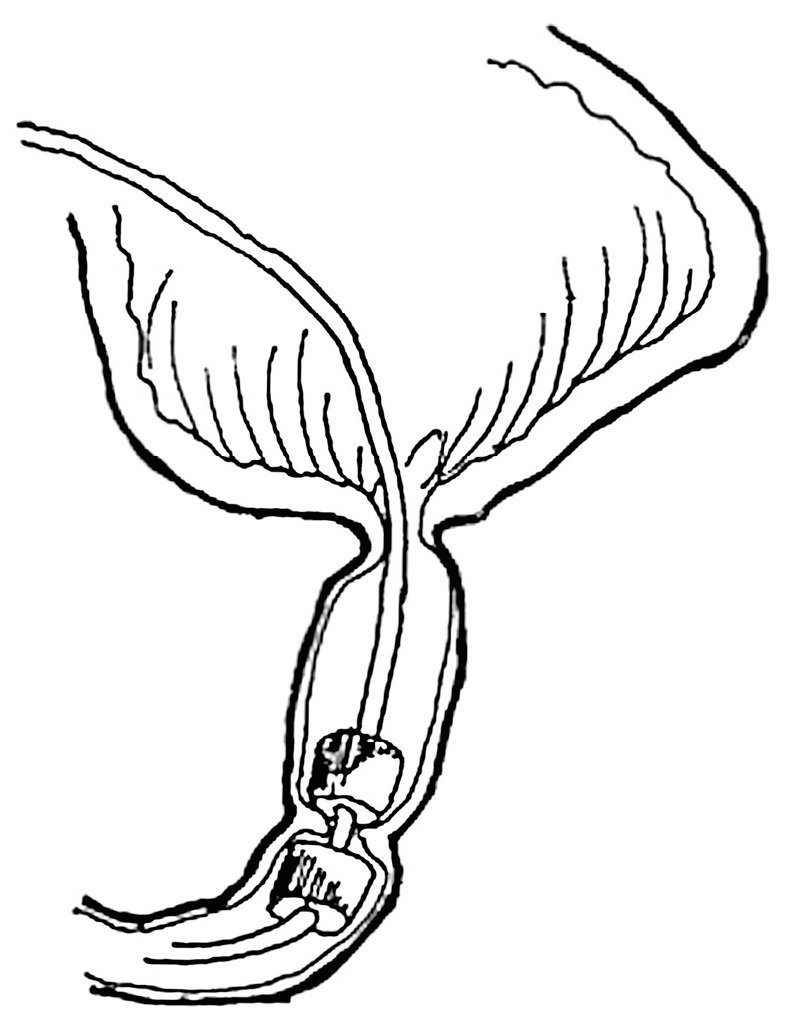
Schematic drawing of endoluminal urethral recanalization using magnetic cylinders.

**Figure 7 children-10-01328-f007:**
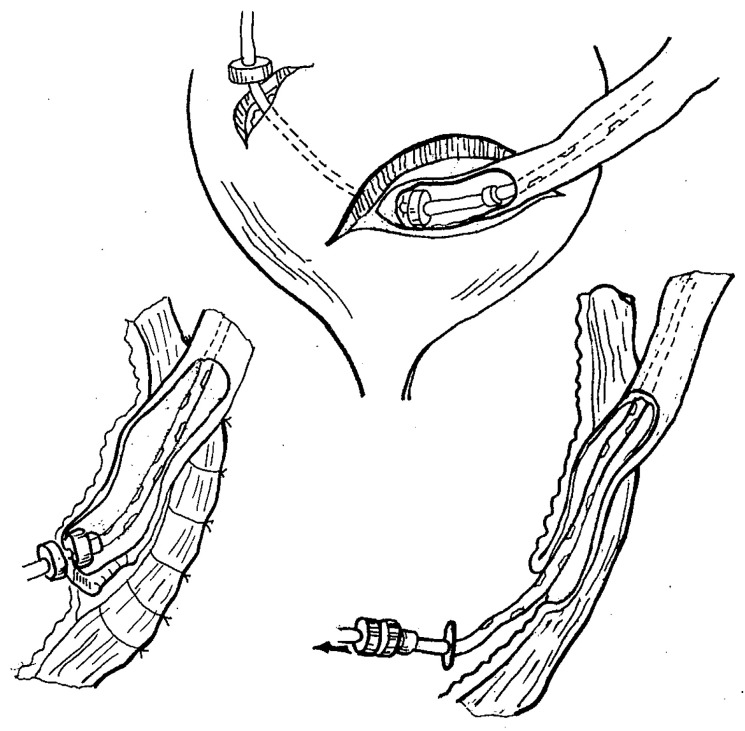
Principle of extravesical ureterocystoneostomy using magnetic rings.

**Table 1 children-10-01328-t001:** Overview of preclinical MCA experiments.

MCA Locations and Types	Animal Model	*n*
Small intestine (end-to-end and side-to-side)	rabbit	96 (42 and 54)
Colorectal end-to-end	canine	47
Ureterovesical side-to-side with Lich-Gregoir tunnelling	canine	55
Urethral end-to-end	canine	52

**Table 2 children-10-01328-t002:** Summary of MCA clinical outcomes.

Indications for MCA	Successful Outcomes	Adverse Outcomes
Esophageal recanalization (*n* = 15)	Esophageal patency restored (*n* = 9)	Perforation (*n* = 1) Restenosis (*n* = 5)
Ileostomy undiversion (*n* = 46)	Intestinal passage restored (*n* = 44)	Technical failure (*n* = 2)
Swenson pull-through (*n* = 10)	Colorectal junction patent (*n* = 6)	Postop stenosis (*n* = 2)
Urethral recanalization (*n* = 5)	Urethral patency restored (*n* = 4)	Partial restenosis (*n* = 1)
Extravesical ureterocystostomy (*n* = 11)	Neo-orifice functional (*n* = 11)	none
**Total (*n* = 87)**	***n* = 76, overall success rate 87.3%**	***n* = 11, adverse outcomes in 12.6%**

## Data Availability

Data on which this publication is based are available from the corresponding author O.J.M. upon reasonable request.

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
