# Peer review of "Magnetic Compression Anastomosis–Past Experience and Current Proposals for Further Development in Pediatric Minimally Invasive Surgery"

_children, 2023, doi:10.3390/children10081328_

Round 1
Reviewer 1 Report
This study is a retrospective report on Magnetic Compression Anastomosis. The report is well-written, but there are some concerns.
-
The data for this review is derived from only one institutional experience. Please 'review' other studies on magnetic compression anastomosis.
-
Why did the author continue using this procedure since 1995? Are there any serious concerns about this technique?
Author Response
See responses to your critique highlighted in yellow in the attached document.
Thank you very much for your valuable comments.

Reviewer 2 Report
Dear Authors,
The content of the paper is not interesting and touch a topic in which there are a lot of data in literature.
Unfortunately, your paper present structural problems related to the data presented.
Quite all cited papers are dated between 1980-1995 and at the moment doesn’t add nothing to the literature. Do you perform preclinical experiments on dog? Why not animals commonly used for preclinical experiments such as pig? Why I should perform a magnetic colo-rectal or small intestine anastomosis?
Currently there are a lot of study published on magnetic anastomosis on child. Maybe you could perform a revision of these study.
No comment about it
Author Response

(The authors gave the same response as above.)

Reviewer 3 Report
This study reports the early experience on MCA performed in Moscow between the 1980-1990s. The study reports data from animal and human studies on the use of MCA in several scenarios with optimal outcomes. The initial reports were done in Russian language and were not reported elsewhere due to language barrier. Despite the results are 30-40 years old and the study is based on retrospective review of data, thesis and papers I believe the topic is of interest and the technique is currently under investigation by different manufacturers. The presence of this data provides stronger input to pursue this research path. The review is clear and well written. I would only request one comment. Do the authors believe that MCA is still superior to mechanical sutures or hand-sewn sutures with more modern sutures (not polyglactin or catgut)? Could they provide a small comment on this?
Author Response

(The authors gave the same response as above.)

Round 2
Reviewer 2 Report
The content of the article does not introduce anything new respect to the current literature.
We should rather analyze the results of human clinical trials rather.